# Outer-membrane-acting peptides and lipid II-targeting antibiotics cooperatively kill Gram-negative pathogens

Qian Li[1,3,5], Rubén Cebrián[1,5], Manuel Montalbán-López[1,4], Huan Ren[2], Weihui Wu[2] & Oscar P. Kuipers [1✉]

The development and dissemination of antibiotic-resistant bacterial pathogens is a growing global threat to public health. Novel compounds and/or therapeutic strategies are required to face the challenge posed, in particular, by Gram-negative bacteria. Here we assess the combined effect of potent cell-wall synthesis inhibitors with either natural or synthetic peptides that can act on the outer-membrane. Thus, several linear peptides, either alone or combined with vancomycin or nisin, were tested against selected Gram-negative pathogens, and the best one was improved by further engineering. Finally, peptide D-11 and vancomycin displayed a potent antimicrobial activity at low µM concentrations against a panel of relevant Gram-negative pathogens. This combination was highly active in biological fluids like blood, but was non-hemolytic and non-toxic against cell lines. We conclude that vancomycin and D-11 are safe at >50-fold their MICs. Based on the results obtained, and as a proof of concept for the newly observed synergy, a *Pseudomonas aeruginosa* mouse infection model experiment was also performed, showing a 4 log$_{10}$ reduction of the pathogen after treatment with the combination. This approach offers a potent alternative strategy to fight (drug-resistant) Gram-negative pathogens in humans and mammals.

[1] Department of Molecular Genetics, Groningen Biomolecular Sciences and Biotechnology Institute, University of Groningen, Nijenborgh 7, 9747AG Groningen, The Netherlands. [2] State Key Laboratory of Medicinal Chemical Biology, Key Laboratory of Molecular Microbiology and Technology of the Ministry of Education, Department of Microbiology, College of Life Sciences, Nankai University, 30071 Tianjin, China. [3] Present address: State Key Laboratory of Biocatalysis and Enzyme Engineering, Hubei Collaborative Innovation Center for Green Transformation of Bio-Resources, Hubei Key Laboratory of Industrial Biotechnology, School of Life Sciences, Hubei University, 430062 Wuhan, China. [4] Present address: Department of Microbiology, Faculty of Sciences, University of Granada, Av. Fuentenueva s/n, 18071 Granada, Spain. [5] These authors contributed equally: Qian Li, Rubén Cebrián. ✉email: o.p.kuipers@rug.nl

Antimicrobial resistance (AMR) is a natural process enhanced by the widespread use and misuse of antibiotics. The critical increase of AMR poses a major and serious worldwide threat to public health. Following the introduction of lipopeptides in 2003, hardly any new families of antibiotics have been developed, while the existing drugs are rapidly becoming ineffective[1–4]. Currently, 700,000 deaths per year can be attributed to the spread of antimicrobial resistance and this number is estimated to increase to 10,000,000 by the year 2050[5]. In 2017, the World Health Organization (WHO) published the global priority list, with 12 antibiotic-resistant bacteria for which new antimicrobials or therapeutic strategies are urgently needed[6]. Remarkably, 9 of the 12 "superbugs" listed are Gram-negative pathogens. Their outer-membrane acts as an efficient barrier to prevent various antimicrobials from reaching their targets at the inner membrane and/or the cytoplasm, which complicates the development of efficient treatments against (multidrug-resistant, MDR) bacteria[7,8]. Passage of the outer-membrane is an essential step in the killing of Gram-negative cells by many antibiotics and antimicrobials. The efficacy of individual antimicrobials can be enhanced by combination with other (either antimicrobial or inactive) permeabilizing compounds[9,10].

Cell-wall biosynthesis and maintenance are essential processes in bacteria and their inhibition leads to growth arrestment and death. In fact, diverse groups of successful antibiotics abolish cell-wall biosynthesis as their primary mechanism of action. Lipid II is a key and highly conserved molecule bringing the cell-wall building blocks to the outside of the cellular membrane[11]. Nisin and vancomycin are potent antimicrobials inhibiting cell-wall biosynthesis by binding to lipid II at the cell membrane using two different binding motifs. They have low micromolar minimal inhibitory concentrations (MICs) but are poorly active on Gram-negative bacteria due to the presence of their outer-membrane[12–14]. Recently, the replacement of the C terminus of the lantibiotic nisin with short peptide sequences increased the antimicrobial activity 4- to 12-fold against relevant Gram-negative microorganisms[15], showing the outer-membrane permeating potency of these peptides. Thus, we hypothesized that if we could find compounds that can perturb or transiently permeate the Gram-negative outer-membrane, this would enable vancomycin or nisin to access the periplasm and reach lipid II, resulting in substantial activity against Gram-negative pathogens[16].

In this study, eight, either natural or synthetic, peptides ranging in size between 11 and 23 amino acids, which exert low to modest activity against Gram-negative bacteria, were selected from literature to work as possible Gram-negative outer-membrane-penetrating or -perturbing peptides, thereby exposing lipid II in the inner membrane to either nisin or vancomycin. A first screening was done using nisin or vancomycin as a partner in combination with either peptide. Nine additional variants of the best peptide identified, i.e., L-8, were designed and the best candidate was used to potentiate the activity of the anti-Gram-positive antibiotic vancomycin. Eventually, the D-11 peptide was obtained, which showed very high synergistic activity when combined with vancomycin. We demonstrate its safety with respect to cytotoxicity, hemolysis, and activity in vivo, indicating its high potential in combination with vancomycin as a future therapeutic agent.

## Results

**Minimal inhibitory concentration (MICs) of peptides alone and in combination with lipid II-targeting antimicrobials.** Vancomycin and nisin were selected because they are potent antimicrobials against Gram-positive bacteria whereas they display low or absent activity against Gram-negative bacteria. Also, they are considered as safe antimicrobials in food preservation (nisin)[17] or in clinical application (vancomycin)[14], and both use the same target, lipid II, to kill bacteria. Eight peptides were selected from literature that displayed low or modest activity against Gram-negative bacteria. Initially, activity tests for vancomycin, nisin and the selected peptides were performed against five clinically relevant Gram-negative pathogens (Table 1). Their MICs ranged from 0.5 μM to more than 256 μM in a strain-dependent manner and, as expected, the antimicrobial activity of either vancomycin or nisin was quite low (Table 1).

After this test, the peptides L-1 to L-8 were combined with either vancomycin or nisin and initially tested against *E. coli*

**Table 1 Sequences and MIC (μM) of the different peptides and antimicrobials used in this work. D amino acids are indicated in lowercase.**

| Name | Sequence | Source | E. coli LMG 15862 | K. pneumoniae LMG 20218 | P. aeruginosa LMG 6395 | A. baumannii LMG 01041 | E. aerogenes LMG 02094 |
|---|---|---|---|---|---|---|---|
| L-1 | GNNRPVYIPQPRPPHPRL | 62 | 0.5 | 5 | >64 | >64 | 8 |
| L-2 | VDKPPYLPRPRPPRRIYNR | 63 | 4 | 8 | >64 | 16 | >64 |
| L-3 | VDKGSYLPRPTPPRPIYNRN-NH$_2$ | 64 | 3 | 16 | 32 | 8 | >32 |
| L-4 | RLLFRKIRRLKR | 65 | 64 | >256 | >128 | >128 | >256 |
| L-5 | RIWVIWRR-NH$_2$ | 66 | 3 | 6 | 3 | 2 | 5 |
| L-6 | RRLFRRILRWL-NH$_2$ | 67 | 2 | 4 | 3 | 2 | 8 |
| L-7 | GIGKHVGKALKGLKGLLKGLGEC | 68 | 6 | 12 | 13 | 2 | 8 |
| L-8 | KRIVQRIKKWLR-NH$_2$ | 69 | 4 | 64 | 16 | 16 | 32 |
| L-9 | -RIVQRIKKW-R-NH2 | This work | 256 | >256 | >256 | >256 | >256 |
| L-10 | -RIVQRIKKWL--NH2 | This work | 32 | 64 | 16 | 128 | >256 |
| L-11 | -RIVQRIKKWLR-NH2 | This work | 12 | 32 | 16 | >64 | 128 |
| L-12 | -KIVQRIKKWLR-NH2 | This work | 16 | 32 | 16 | 128 | 256 |
| L-13 | -AIVQRIKKWLR-NH2 | This work | 12 | 24 | 16 | 128 | 128 |
| L-14 | -RIRKRIKKWLR-NH2 | This work | 16 | >256 | 16 | >256 | >256 |
| L-15 | -RIKRRIKKWLR-NH2 | This work | 24 | >256 | 32 | >256 | >256 |
| D-11 | rivqrikkwlr-NH2 | This work | 4 | 32 | 16 | 8 | 32 |
| D-11R | rlwkkirqvir-NH2 | This work | 8 | 32 | 16 | 64 | 32 |
| Vancomycin | | | 64 | 128 | 128 | 32 | 192 |
| Nisin | | | 12 | 48 | 36 | 6 | 32 |

**Table 2 Combined activity of L-1 to L-8 peptides and either vancomycin or nisin against five Gram-negative pathogens.**

| | L-1 | L-2 | L-3 | L-4 | L-5 | L-6 | L-7 | L-8 |
|---|---|---|---|---|---|---|---|---|
| *E. coli* LMG 15862 | | | | | | | | |
| Vancomycin | 32 | 32 | 32 | 16 | 32 | 16 | 16 | 32 |
| Peptide | 0.125 | 2 | 1.5 | 16 | 0.75 | 1 | 3 | 0.25 |
| FICI | 0.75 | 1 | 1 | 0.75 | 0.75 | 0.75 | 0.75 | 0.563 |
| Nisin | 1.5 | 1.5 | 3 | 3 | 0.75 | 1.5 | 1.5 | 1.5 |
| Peptide | 0.13 | 2 | 1.5 | 8 | 0.75 | 0.5 | 1.5 | 1 |
| FICI | **0.375** | 0.625 | 0.75 | **0.375** | **0.313** | **0.375** | **0.375** | **0.375** |

| | L-1 | L-5 | L-6 | L-7 | L-8 |
|---|---|---|---|---|---|
| *P. aeruginosa* LMG 6395 | | | | | |
| Vancomycin | 128 | 128 | 32 | 128 | 64 |
| Peptide | 32 | 1.5 | 1.5 | 6.5 | 8 |
| FICI | 1.25 | 1.5 | 0.75 | 1.5 | 1 |
| Nisin | 4.5 | 2.25 | 4.5 | 4.5 | 4.5 |
| Peptide | 64 | 0.75 | 0.75 | 3.25 | 1 |
| FICI | 0.625 | **0.313** | **0.375** | **0.375** | **0.188** |
| *K. pneumoniae* LMG 20218 | | | | | |
| Vancomycin | 64 | 64 | 32 | 64 | 8 |
| Peptide | 0.625 | 3 | 1 | 3 | 8 |
| FICI | 0.625 | 1 | **0.5** | 0.75 | **0.188** |
| Nisin | 6 | 6 | 3 | 3 | 0.75 |
| Peptide | 1.25 | 1.5 | 0.5 | 0.75 | 4 |
| FICI | **0.375** | **0.375** | **0.188** | **0.125** | **0.078** |
| *A. baumannii* LMG 01041 | | | | | |
| Vancomycin | 16 | 16 | 8 | 16 | 4 |
| Peptide | 16 | 0.5 | 0.5 | 0.25 | 4 |
| FICI | 0.625 | 0.75 | **0.5** | 0.625 | **0.375** |
| Nisin | 1.5 | 0.75 | 0.75 | 0.75 | 0.19 |
| Peptide | 64 | 0.5 | 0.5 | 0.5 | 1 |
| FICI | 0.546 | **0.375** | **0.375** | **0.375** | **0.094** |
| *E. aerogenes* LMG 02094 | | | | | |
| Vancomycin | 192 | 192 | 64 | 192 | 48 |
| Peptide | 8 | 5 | 2 | 4 | 4 |
| FICI | 2 | 2 | 0.583 | 1.5 | **0.375** |
| Nisin | 4 | 4 | 2 | 4 | 2 |
| Peptide | 4 | 1.25 | 1 | 1 | 8 |
| FICI | 0.625 | **0.375** | **0.188** | **0.25** | **0.313** |

In the table are listed the MICs (μM) for vancomycin, nisin and the peptides for the best combination, as well as the FICI calculation. For an explanation of FICI and how it is calculated see the "Methods" section. Synergistic combinations are labeled in bold. For the FICI calculations twice the highest concentration tested was used in the cases where the MIC was not reached. Individual MICs are listed in Table 1.

LMG 15862. Interestingly, all the peptides tested displayed some synergistic effect with nisin against *E. coli*, with the exception of L-2 and L-3 that only showed an additive effect (Table 2). However, in the combination with vancomycin, the effect was mainly additive and L-8, with a Fractional Inhibitory Concentration Index (FICI) of 0.563, showed the best synergy (Table 2). Considering the best reduction in the MIC of either vancomycin or nisin, the peptides L-1, L-5, L-6, L-7 and L-8 were selected for further characterization of the synergistic effect against a panel of 4 other Gram-negative medically relevant bacteria (Table 2).

The combination of the selected peptides with nisin was synergistic in all the cases against all the bacteria tested with the exception of the L-1 peptide (Table 2). The combination of L-8 with either vancomycin or nisin displayed the best synergism against almost all of the five Gram-negative pathogens, while the combined effect of the other peptides with vancomycin was modest and, in general, additive, and strain-specific (Table 2). The peptide L-6, which was also (less) synergistic in some combinations, was discarded because it showed rather high hemolytic activity in the previous studies[18]. Notably, L-8 had very low hemolytic activity[19], which made it the best candidate for

further studies. Based on these results, peptide L-8 was selected for a round of further modifications.

**L-8 peptide improvement.** In order to increase the combined effect, seven variants of L-8 were designed. Several parameters such as peptide truncation, role of arginine/lysine, amphiphilicity, and hydrophobicity were explored[20]. Based on this, three shorter variants were designed to increase the drug potential of the peptide and then, for the best one, another four variants were designed (Table 1 and Supplementary Fig. 1). In general, none of the newly designed peptides systematically outperformed L-8 (Table 1). However, in combination with either vancomycin or nisin an astonishing improvement of the synergy was observed for the peptide L-11 (Table 3). This peptide, being just one amino acid smaller than the original L-8, was better in synergy with vancomycin in all the cases against all the bacteria as well as in the combination with nisin, with the exception of *Pseudomonas*, where the FICI value was the same (Table 3). The reduction in the MIC of vancomycin in combination with L-11 ranged between 8-fold reduction for *Pseudomonas* and 32-fold for *Klebsiella*. The synergistic effect varied in a strain-specific manner with the other designed peptides (Table 3). Considering several parameters as

**Table 3 Combined activity of L-peptides and either vancomycin or nisin against five Gram-negative pathogens.**

|  | L-8 | L-9 | L-10 | L-11 | L-12 | L-13 | L-14 | L-15 |
|---|---|---|---|---|---|---|---|---|
| *E. coli* LMG 15862 |  |  |  |  |  |  |  |  |
| Vancomycin | 32 | 8 | 8 | 4 | 4 | 4 | 4 | 4 |
| Peptide | 0.25 | 16 | 4 | 1.5 | 2 | 3 | 2 | 3 |
| FICI | 0.563 | **0.188** | **0.25** | **0.188** | **0.188** | **0.313** | **0.188** | **0.188** |
| Nisin | 1.5 | 6 | 3 | 1.5 | 1.5 | 1.5 | 3 | 3 |
| Peptide | 1 | 64 | 2 | 1.5 | 2 | 1.5 | 2 | 3 |
| FICI | 0.375 | 0.75 | **0.313** | **0.25** | **0.25** | **0.25** | 0.375 | 0.375 |
| *P. aeruginosa* LMG 6395 |  |  |  |  |  |  |  |  |
| Vancomycin | 64 | ND | 16 | 16 | 16 | 4 | 4 | 16 |
| Peptide | 8 | ND | 2 | 2 | 2 | 8 | 8 | 8 |
| FICI | 1 | ND | **0.25** | **0.25** | **0.25** | **0.531** | **0.531** | **0.5** |
| Nisin | 4.5 | ND | 4.5 | 4.5 | 4.5 | 4.5 | 2.25 | 9 |
| Peptide | 1 | ND | 4 | 1 | 4 | 8 | 4 | 8 |
| FICI | 0.188 | ND | 0.375 | 0.188 | 0.375 | 0.625 | 0.313 | 0.5 |
| *K. pneumoniae* LMG 20218 |  |  |  |  |  |  |  |  |
| Vancomycin | 8 | ND | 16 | 4 | 16 | 16 | ND | ND |
| Peptide | 8 | ND | 16 | 2 | 8 | 3 | ND | ND |
| FICI | 0.188 | ND | 0.375 | **0.094** | 0.375 | 0.25 | ND | ND |
| Nisin | 0.75 | ND | 3 | 0.75 | 0.75 | 1.5 | ND | ND |
| Peptide | 4 | ND | 1 | 1 | 2 | 3 | ND | ND |
| FICI | 0.078 | ND | 0.078 | **0.047** | 0.078 | 0.156 | ND | ND |
| *A. baumannii* LMG 01041 |  |  |  |  |  |  |  |  |
| Vancomycin | 4 | ND | 1 | 2 | 4 | 4 | ND | ND |
| Peptide | 4 | ND | 16 | 1 | 4 | 8 | ND | ND |
| FICI | 0.375 | ND | 0.125 | **0.070** | **0.156** | **0.188** | ND | ND |
| Nisin | 0.19 | ND | 0.375 | 0.19 | 0.375 | 0.375 | ND | ND |
| Peptide | 1 | ND | 16 | 4 | 16 | 32 | ND | ND |
| FICI | 0.094 | ND | 0.188 | **0.070** | 0.188 | 0.313 | ND | ND |
| *E. aerogenes* LMG 02094 |  |  |  |  |  |  |  |  |
| Vancomycin | 48 | ND | ND | 12 | 12 | 12 | ND | ND |
| Peptide | 4 | ND | ND | 8 | 16 | 16 | ND | ND |
| FICI | 0.375 | ND | ND | **0.125** | **0.125** | **0.188** | ND | ND |
| Nisin | 2 | ND | ND | 4 | 4 | 4 | ND | ND |
| Peptide | 8 | ND | ND | 8 | 1 | 8 | ND | ND |
| FICI | 0.313 | ND | ND | **0.188** | **0.128** | **0.188** | ND | ND |

In this table, the MICs (μM) of vancomycin, nisin, and the peptides for the best combination, as well as the FICI calculation, are listed. Peptides with a better synergism coefficient than L-8 are displayed in bold. For the FICI calculations, twice the highest concentration tested was used in the cases where the MIC was not reached. Individual MICs are listed in Table 1.
*ND* no synergy determined.

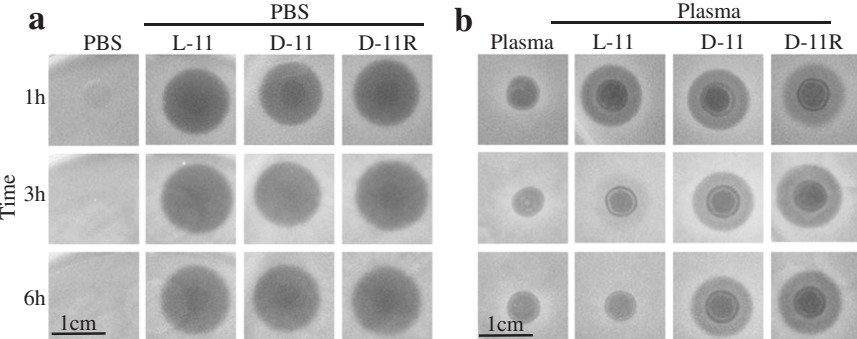

**Fig. 1 Plasma stability for the peptides L-11, D-11 and D-11R. a** Antimicrobial activity of the peptides L-11, D-11, and D-11R against *K. pneumoniae* LMG 20218 after their incubation in PBS. **b** Antimicrobial activity of the peptides L-11, D-11, and D-11R after their incubation in human plasma.

size, activity alone and activity in the combinations, peptide L-11 was selected for further (pre-clinical) characterization.

**Plasma stability.** In view of the clinical potential, plasma stability is a desirable quality of newly designed antimicrobials[21,22]. So, before further analysis, the stability of L-11 in plasma was analyzed. The peptide was incubated in PBS (Fig. 1a) and plasma (Fig. 1b), and the reduction in the antimicrobial activity was tested by a spot-on-lawn assay against *K. pneumoniae*. The peptide L-11 was not very stable in plasma, being almost completely degraded within the first 3 h (Fig. 1b). In order to increase the stability, the peptide was synthesized with only D-amino acids

(D-11) making it resistant against peptidases and proteases[23]. Moreover, reversed peptides (with a reversal of the peptide sequence) have been reported to be interesting in peptidomimetics and can display good activity[18,24]. Thus, both variants were synthesized and assayed (D-11 and D-11R). As we can see in Fig. 1b, both D-11, as well as its reversed peptide D-11R, were completely resistant to degradation in plasma.

**Antimicrobial activity and synergy of the newly designed peptides.** The introduction of D-amino acids or the change in the order of the amino acids may abolish a (putative) receptor interaction, thus allowing to establish whether or not a specific

(proteinaceous) receptor is playing a role. For this reason, and to determine whether the changes affected the previously observed activity, the new variants were tested alone and in synergy with vancomycin against pathogenic multiresistant bacteria, including clinical isolates (Table 1). The impact of D-amino acid replacement in L-11 entailed a drastic improvement against *E. aerogenes* and *A. baumannii* (4 and >8-fold MIC reduction, respectively, for the peptide alone), although no important changes against the rest of the panel was observed. The reversed D-peptide was in general either equally or 2- to 8-fold less active than the D-peptide against the whole panel (Table 1). D-11 and D-11R showed, just as L-11, a strong synergistic effect in all the combinations with either vancomycin or nisin (Supplementary Table 1). Nevertheless, the combinations of D-11 and either vancomycin or nisin showed slightly less synergism than the L-11 peptide with a FICI value ranging between 0.078 and 0.25 but a similar reduction in the MIC value for vancomycin or nisin, so these values still indicate a very strong in vitro synergy (Supplementary Table 1). In general, the combination with nisin was more effective and yielded the lowest FICI values. Notably, the D-peptides exhibited an enhanced activity against Gram-negative pathogens, when compared to the L-counterpart. The reversed peptide sequence did not largely change the activities of the peptides except against *A. baumannii*. These results suggest that the selected L-11, D-11, and D-11R do not interact with a specific proteinaceous receptor, which would require stereospecific interactions.

Since D-amino acids confer higher resistance against proteolysis, are more stable in serum and blood, and because the activity in synergy was not greatly altered, D-11 was selected for further tests in combination with vancomycin against clinical isolated MDR bacteria. Synergy with vancomycin may constitute an interesting therapeutic alternative, since it is smaller than nisin and -in contrast to nisin- is currently used as an antibiotic in clinical therapy. As before (Table 1), the vancomycin activity alone against these strains was low, while the combined activity of vancomycin and D-11 was strain-specific and synergistic (Table 4). The most sensitive strains to vancomycin in combination with D-11 were *E. coli* and *A. baumannii*, and the best synergistic effect was obtained for the strains *E. coli* ATCC BAA-2452, *K. pneumoniae* ATCC BAA-2524, *K. pneumoniae* B1945, and *A. baumannii* ATCC BAA-1605 (Table 4).

The bactericidal effect of vancomycin, D-11, and the combinations thereof were determined against five Gram-negative strains. The final concentration of the compounds used was 10-fold the MIC of each compound measured either individually (Table 1) or in combination (Table 3). Unlike the positive controls (untreated cells), no colonies of any of the five Gram-negative pathogens could grow after treatment incubation for 3 h (Supplementary Fig. 2). There was no growth recovery after exposure to vancomycin, D-11, and the combinations thereof, so both the compounds alone and the combinations thereof are bactericidal.

**Mechanistic aspects of the synergistic action.** According to our hypothesis, the peptide is perturbing the outer-membrane, enabling vancomycin to access the periplasm, thus reaching its target lipid II. To test whether the outer-membrane is the main target of D-11, the antimicrobial activity of D-11 and vancomycin was tested against 5 different Gram-positive bacteria, which do not have an outer-membrane (Supplementary Table 2). D-11 MIC was above 64 μM in all cases (with the exception of *E. faecium* that was 10.4 μM). In addition, no synergy in the combination with vancomycin was observed. This data suggests that the activity of D-11 is due to its action on the outer-membrane.

*The role of LPS and Mg$^{2+}$ in the activity.* To prove the role of the outer-membrane structure in the synergistic killing mechanism, we tested the influence of LPS and magnesium ions on the activity of vancomycin and D-11 against the Gram-negative bacteria panel. Both vancomycin and D-11 (especially this one) exhibited a reduced antimicrobial effect against all the tested pathogens when the cells were grown in the presence of exogenous LPS or Mg$^{2+}$ (Supplementary Table 3). When LPS was in the medium, the MICs of vancomycin against Gram-negative pathogens increased 2-fold, except against *A. baumannii*. Remarkably, 2- to more than 16-fold D-11 was needed to inhibit the growth of Gram-negative pathogens in the presence of LPS and this reduction was even more drastic in the presence of an outer-membrane-stabilizing agent as Mg$^{2+}$ (Supplementary Table 3).

The effect of exogenous LPS and Mg$^{2+}$ was also analyzed in the synergistic combination. As depicted in Fig. 2a, LPS considerably suppressed the ability of D-11 to potentiate vancomycin against Gram-negative pathogens. As an example, 2 μM vancomycin and 1 μM D-11 can inhibit the growth of *K. pneumoniae* without the addition of LPS. However, at least 32 μM vancomycin and 16 μM D-11 were needed after the addition of excess LPS. When 21 mM Mg$^{2+}$ was added to the growth medium for the checkerboard

---

**Table 4 Combined activity of D-11 and vancomycin (μM) against an extended panel of MDR Gram-negative clinical isolated pathogens.**

| Strains | MIC$_A$ | MIC$_B$ | MIC$_{AB}$ | MIC$_{AC}$ | FICI |
|---|---|---|---|---|---|
| *E. coli* ATCC 25922 | >88.32 | 5.36 | 5.52 | 1.6 | **0.329** |
| *E. coli* ATCC BAA-2452 | >88.32 | 5.36 | 2.76 | 0.80 | **0.164** |
| *E. coli* B1927 | 88.32 | 2.68 | 5.52 | 1.6 | 0.659 |
| *K. pneumoniae* ATCC700603 | >88.32 | >85.68 | 11.04 | 10.71 | **0.081** |
| *K. pneumoniae* ATCC BAA-2524 | >88.32 | >85.68 | 11.04 | 3.21 | **0.081** |
| *K. pneumoniae* B1945 | >88.32 | 21.41 | 11.04 | 3.21 | **0.212** |
| *P. aeruginosa* ATCC 27853 | >88.32 | 42.84 | 11.04 | 10.71 | **0.312** |
| *P. aeruginosa* ATCC BAA-2108 | >88.32 | >85.68 | 44.15 | 42.84 | **0.499** |
| *P. aeruginosa* B1954 | 88.32 | 85.68 | 44.15 | 42.84 | 1 |
| *P. aeruginosa* PA14 | 88.32 | 128 | 32 | 8 | **0.422** |
| *A. baumannii* ATCC 17978 | 88.32 | 10.71 | 5.52 | 1.61 | **0.22** |
| *A. baumannii* ATCC BAA-1605 | 44.16 | 10.71 | 2.76 | 0.80 | **0.137** |
| *A. baumannii* B2026 | 22.08 | 10.71 | 2.76 | 2.68 | **0.376** |

MIC$_A$ is the MIC of vancomycin alone; MIC$_B$ corresponds to the MIC of D-11 when used alone; MIC$_{AB}$ is the MIC of vancomycin in combination with the peptide at the MIC$_{AC}$ concentration. MIC$_{AC}$ is the MIC of D-11 when used with the MIC$_{AB}$ concentration of vancomycin. Synergistic combinations are labeled in bold. For the FICI calculations, twice the highest concentration tested was used in the cases where the MIC was not reached.

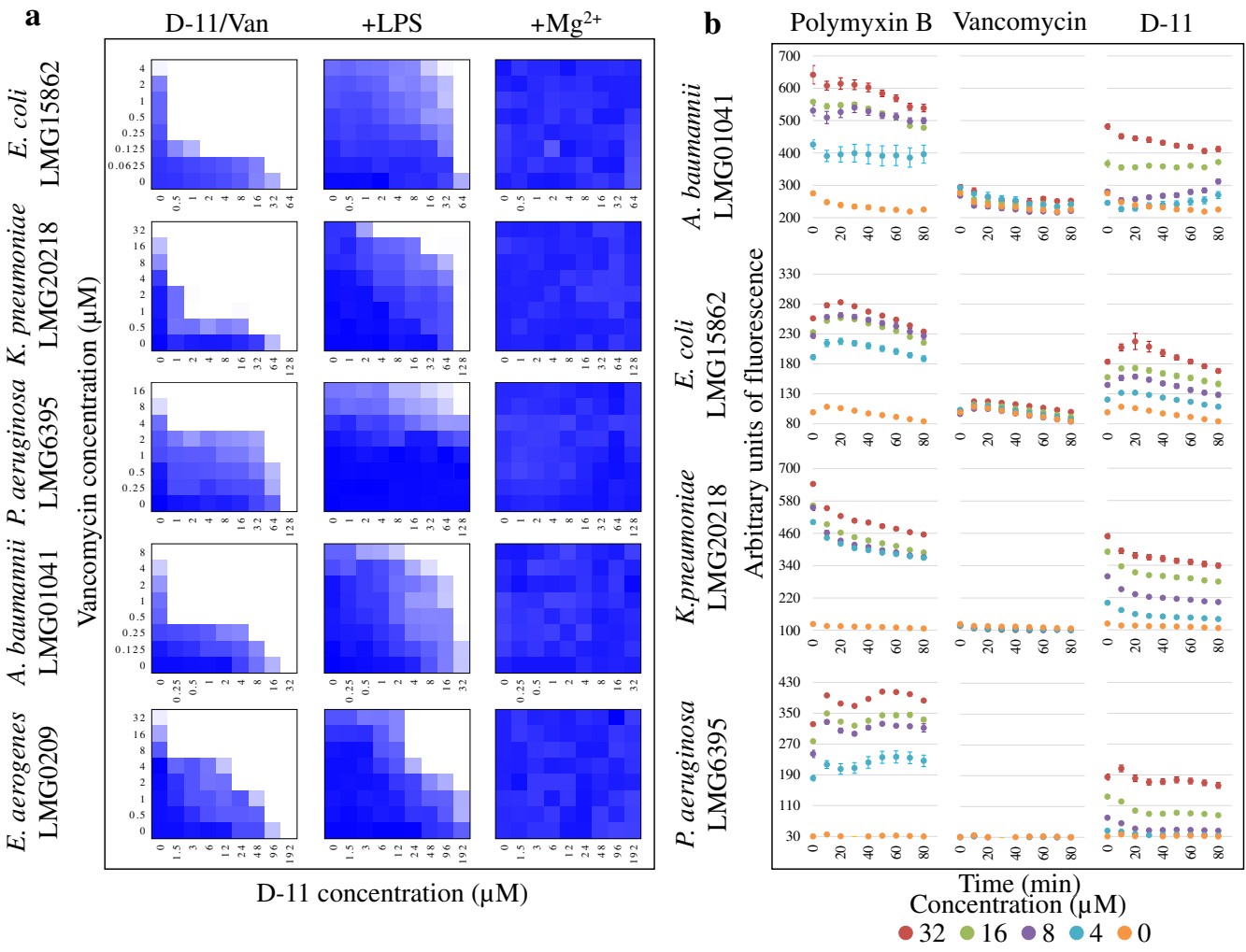

**Fig. 2 Effect of LPS in the D-11/vancomycin synergism and outer-membrane permeabilization assay. a** Checkerboard broth microdilution assay diagram showing the effect of LPS and Mg$^{2+}$ in the D-11/vancomycin synergistic relation. Dark region represents higher cell density. **b** Dose-dependent outer-membrane permeabilization by D-11 against different Gram-negative bacteria. Polymyxin B was used as a positive control and vancomycin as negative.

broth microdilution assays, D-11 was not able to potentiate vancomycin against any of the five Gram-negative pathogens, which grew at the tested concentrations (Fig. 2a). This is consistent with the results when vancomycin and D-11 were tested alone in the presence of LPS and Mg$^{2+}$ (Supplementary Table 3) and suggests that the outer-membrane is the main target for D-11.

*D-11 is perturbing the outer-membrane.* The activity of D-11 on the outer-membrane was tested measuring NPN (1-*N*-phenylnaphthylamine) uptake. The outer-membrane was permeabilized by D-11 in a dose-depending-manner in all the Gram-negative bacteria tested (Fig. 2b). As expected, no permeabilization was observed for vancomycin, while polymyxin B (positive control) exerted a strong activity. These results suggest that the D-11 peptide can disrupt the integrity of the outer-membrane of these Gram-negative pathogens via association with LPS, thereby enhancing the permeability of the outer-membrane. This helps to sensitize the Gram-negative pathogens to antibiotics as vancomycin that, otherwise, are restricted to act against Gram-positive bacteria.

**Cell toxicity and hemolytic activity of vancomycin and D-11.** In order to prove the safety of vancomycin, D-11, and their

combinations, toxicity tests against human cell lines and hRBCs were performed. As we can see in Fig. 3a, vancomycin and D-11 did not affect the viability of human HEK-293 cells, neither alone nor in combination at all the concentrations tested (up to 88 µM). After that, fresh hRBCs were used for the hemolytic tests. Vancomycin and D-11 did not cause any hemolysis of hRBCs even at concentrations as high as 500 and 600 µM, respectively (Supplementary Table 4).

When combined, vancomycin and D-11 did not cause hemolysis at the highest concentration tested. The cell selectivity of the peptides is identified as the therapeutic index (TI)[19,25], which is a parameter related to the toxicity and the effect of drugs (Supplementary Table 4). A higher TI is preferable for an antimicrobial to be considered safe. The TI of vancomycin and D-11 was 11.4 and 39.03, respectively. The TI of D-11 is ~3.4-fold higher than that of vancomycin and this highlights the safety of peptide D-11.

**Antimicrobial activity in blood.** The stability of D-11 was previously confirmed in plasma. However, whole blood is a complex medium in which the synergistic interaction between vancomycin and D-11 can be affected by many additional parameters and factors. An in vitro bacteremia infection model was developed by the infection of whole blood with 10$^8$ colony forming units

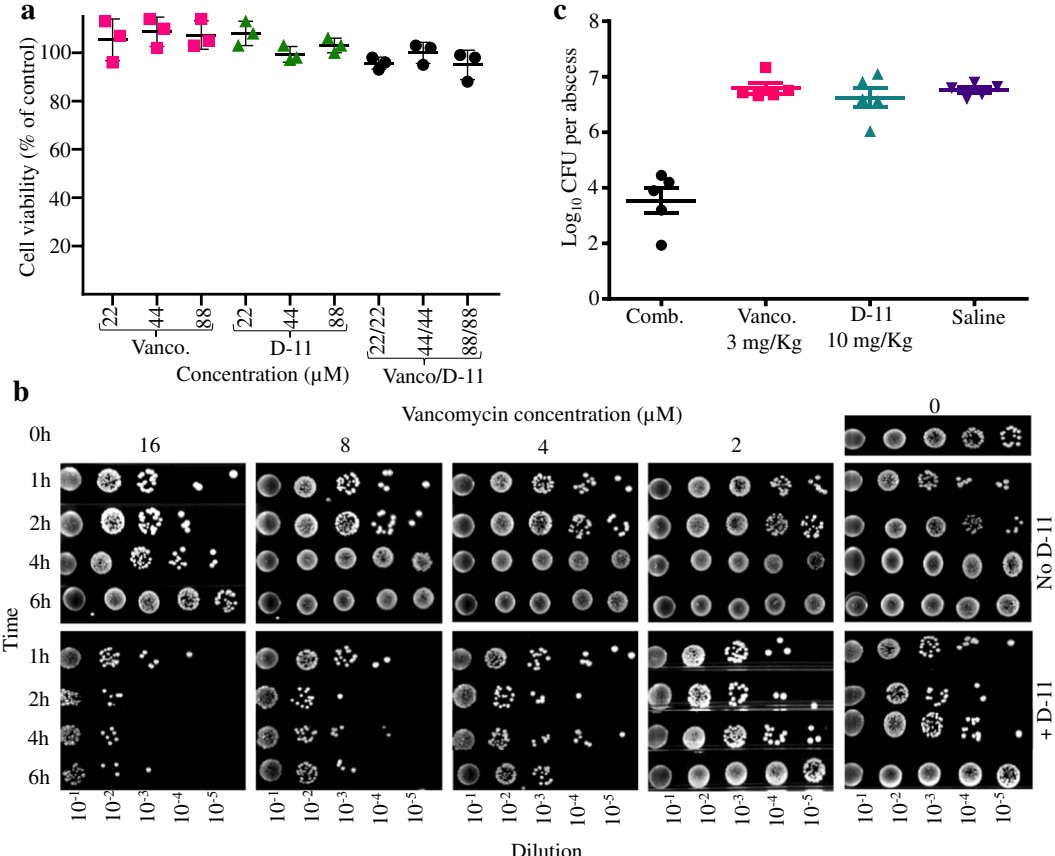

**Fig. 3 Toxicity and activity ex vivo/in vivo of D-11 with or without vancomycin. a** Effect on the cell viability for vancomycin (vanco.), D-11, and their combination against HEK-293 cells. **b** Ex vivo bacteremia model for vancomycin, D-11, and their combinations. 10 µL of serial decimal dilutions of each one of the combinations of vancomycin (16, 8, 4, and 2 µM) with/without D-11 (4 µM) tested in blood were dropped. **c** In vivo efficacy of vancomycin and D-11 alone and its combination (comb.) against *P. aeruginosa* PA14. Saline is the negative control.

(CFU)/mL of *K. pneumoniae* LMG 20218. 10 µL of each one of the tubes were collected at different times, they were decimal serially diluted and plated to quantify the evolution of the infection in blood. As we can see in Fig. 3b, in the presence of vancomycin and D-11, the number of colonies observed was always lower than that of the controls (vancomycin alone or D-11 alone) and during the different time points, with the exception of the combination of 2 µM vancomycin and 4 µM D-11. After 6 h of incubation, $10^{12}$ CFU/mL were counted for the negative controls as well as for the treatment with 8, 4, and 2 µM vancomycin ($10^{10}$ CFU/mL for the treatment with 16 µM vancomycin). In the D-11 treated blood, $10^{12}$ CFU/mL were counted for the negative control (just D-11), $10^{10}$ CFU/mL for the combination with 2 µM vancomycin, $10^{6}$ CFU/mL for the combination with 4 and 8 µM vancomycin and $10^{4}$ CFU/mL for the combination with 16 µM vancomycin.

**A proof of concept. The in vivo efficacy**. Encouraged by the in vitro results and the potential of dose-sparing drug combinations in therapy, we tested the efficacy of D-11 in combination with vancomycin in a murine abscess model of *P. aeruginosa* PA14 (Fig. 3c). Here, we observed that the combination of D-11 (10 mg/kg) and vancomycin (3 mg/kg) was highly efficacious, causing a more than 4 log $_{10}$ reduction of CFU in the abscess (Fig. 3c), whereas after treatment with the individual compounds the cell count was similar to that of the saline negative control.

## Discussion

Combined antibiotic therapy has been pivotal in the treatment of bacterial infections such as tuberculosis and can prevent resistance appearance[26,27]. In addition, (antimicrobial) peptides constitute a yet unused alternative in antimicrobial therapy in spite of their potential[5]. Here, we attempted to expand the spectrum of lipid II binding antimicrobials such as vancomycin and nisin, so that they can be used against (resistant) Gram-negative bacteria by a synergy strategy using natural or synthetic peptides.

Compounds with different targets on bacterial cells can display very good synergistic effects against pathogens, thereby allowing dose-sparing and reducing their individual toxicity, which has an enormous potential in therapy[10,28,29]. In this work, several peptides from different origins were selected and tested for synergy together with two classes of lipid II binding antimicrobials (vancomycin and nisin). Additionally, engineering of the best hits provided a variant, L-11, with great potential to extend vancomycin's spectrum against (MDR) Gram-negative pathogens. Synergies of vancomycin with other antimicrobials are well documented when targeting Gram-positive bacteria[30–32], as well as its synergy in combination with outer-membrane perturbing (or not[33]) antibiotics as colistins[34,35]. However, these combinations are still under study because of the relatively high toxicity observed[36]. There is not much information available on the synergy of vancomycin and peptides against Gram-negative bacteria and, if there are data, the observed effect is additive and the concentrations of vancomycin needed in these combinations are outside of the clinically relevant range[9,37,38].

Plasma stability is a key factor during the clinical development of a compound and peptides that have a short half-life are unlike to reach clinical application[21,22,39]. It is known that D-amino acids are protected against the action of peptidases[23,40], thus, two total D-amino acids forms were synthesized: the direct one (D-11) and the reversed one (D-11R). These two variants were completely stable in plasma and largely retain the activity against an (MDR) Gram-negative bacteria panel. The fact that both the D- and L-amino acids-containing peptides were similarly active indicates that the activity does not rely on binding to a specific receptor and therefore its ability to interact with the outer-membrane was tested.

The most variable part in the outer-membrane is the LPS. To date, more than 200 different LPS forms have been described just for E. coli (ECODAB database)[41]. LPS is a polyanionic molecule with numerous phosphate groups that generates repulsive forces (due to accumulation of the negative charges) that are screened and bridged by the divalent cations ($Mg^{2+}$ and $Ca^{2+}$) which are known to be crucial for the integrity of the outer-membrane[42]. Our data suggest that this peptide is prone to interact with LPS, causing outer-membrane perturbation, which potentiates the activity of Gram-positive-specific antimicrobials (e.g., vancomycin and nisin) against Gram-negative pathogens, including antibiotic-resistant strains. The presence of $Mg^{2+}$ could influence the MIC, not only because of an increased outer-membrane stability, but also because of its electrostatic competition with D-11 (pI = 12.3). In fact, in the presence of $Mg^{2+}$, a larger peptide concentration is needed to reach a coverage high enough on the membrane surface to permeabilize it[43]. Additionally, NPN permeabilization in the presence of D-11 indicates that the outer-membrane is altered, in a similar way as polymyxin B does. This kind of activity has been previously reported for small α-helical and/or cationic peptides[44–46]. Alteration of the overall structure of the outer-membrane is, in principle, difficult to achieve through simple point mutations but would require a deep change in the Gram-negative bacterial physiology. Thus, a compound acting on this essential component constitutes an important advance in anti-Gram-negative antibiotic therapy.

Another important point addressed in this study is the absence of toxicity against HEK-293 cells and also the absence of hemolytic activity even at high concentrations. This is especially important since D-11 is a cationic peptide, which are usually not toxic against cells lines but they are often hemolytic and poorly stable in plasma[47].

Finally, the combination was tested in a bacteremia model in blood. The activity of exogenous antimicrobial peptides in blood has been referred previously against both Gram-negative and positive bacteria[48,49], but never in synergy tests. The combination of D-11 and vancomycin was highly active in blood, controlling the infection by K. pneumoniae in the conditions tested. Importantly, the concentration of vancomycin used in this study falls within the dose range of vancomycin currently applied in humans in the case of Gram-positive bacterial infections (MDR organisms may require daily doses up to 60 mg/kg, equivalent to ~40 μM, www.drugs.com). The in vitro experiments suggest that (combinations of) vancomycin and D-11 are safe at therapeutic concentrations and that the antimicrobial activity of the synergy takes place efficiently in human fluids like blood. These data encouraged us for an in vivo experiment, since the application of antimicrobial peptides often fails due to their low activity in the presence of plasma[50]. As a proof of concept, mice were infected with P. aeruginosa, and after the treatments, we observed an important reduction in the CFU in the mice treated with the combination, whereas the single components did not show this effect. Similar results have been observed for the combination of the antiparasitic drug pentamidine (inactive against bacteria) with some antibiotics[51]. Pentamidine behaves as a powerful adjuvant sensitizing Gram-negative bacteria to typical anti-Gram-positive antibiotics or even to polymyxins in resistant organisms. Remarkably, the poor combined effect with vancomycin suggests a different mechanism of action that could be even complementary in the treatment of Gram-negative pathogens[51].

In summary, the combinations here described constitute a therapeutic approach to deal with (MDR) Gram-negative pathogens. The efficacy of vancomycin and D-11 further indicates their high potential for clinical use. In spite of the relatively low activity of D-11 alone, the astonishing synergism of D-11 with vancomycin and nisin raises the possibility of D-11 to be registered as an adjuvant instead of an antimicrobial itself. The D-11 peptide fulfills the requirement to be considered as a good drug candidate: high efficacy (in synergy), low toxicity to eukaryotic cells, blood stability, high proteolytic stability, and low cost[20]. So far all the results indicate that D-11 has the potential to become an important drug to combat the emerging resistance problem that the world is facing[52]. Further development is currently ongoing to collect all the pre-clinical data to enable first in human studies.

## Methods

**Bacterial strains, cell lines, and growth conditions**. The bacterial strains used in this study are listed in Supplementary Table 5. Gram-negative bacteria were routinely grown in LB medium (Formedium, Norfolk, UK) and shaking (200 rpm) at 37 °C, while Gram-positive strains were grown statically in M-17 broth (BD Difco, New Jersey, US) plus 0.5% glucose (GM-17) at 37 °C with the exception of Streptococcus strains that were cultured in C + Y medium as described by Aprianto et al.[53]. For solid media agar at 1.5 % final concentration was added.

The human embryonic kidney cell line HEK-293 was obtained from the American type culture collection (ATCC) and cultured in Dulbecco's modified Eagle's medium (DMEM) growth medium supplemented with 1% non-essential amino acids (NEAA) and 10% fetal bovine serum (FBS) (Sigma-Aldrich, Ontario, Canada).

**Peptides, antimicrobials, reagents, and chemicals**. The peptides used in this work were synthesized by Pepscan (Lelystad, The Netherland) (Table 1). For the purification and quantification of nisin, commercial nisin was suspended in 0.05% acetic acid and further purified by HPLC (Agilent 1260 Infinity LC) equipped with a semi-preparative C12 column (Phenomenex 250 × 10 mm). The fractions were collected, tested for activity against L. lactis, and analyzed with MALDI-TOF. The active, fully dehydrated and pure fractions were freeze-dried. The freeze-dried nisin was weighted and dissolved with 0.05% acetic acid[54]. Vancomycin, polymyxin B, $MgCl_2$, lipopolysaccharide (LPS) from Escherichia coli O55:B5, HEPES buffer, and chloral hydrate were purchased from Sigma-Aldrich. Sodium phosphate tablets, DMEM, non-essential amino acids (NEAA) were purchased from ThermoFisher Scientific (Waltham, Massachusetts, USA). CellTiter-glo reagent was purchased from Promega (Leiden, the Netherlands). Human plasma EDTA K2 Mixed-Gender was purchased from Sera Laboratories International, Ltd (West Sussex, United Kingdom).

**MIC determination and synergy test**. MIC tests were performed in triplicate by liquid growth inhibition microdilution assays in sterile polypropylene microtiter plates according to the Clinical and Laboratory Standards Institute (CLSI) guideline for bacteria that grow aerobically[55]. Briefly, the indicator strains were first streaked on the appropriate agar plate, and then single colonies were picked and incubated overnight in liquid broth. The final inoculum of bacteria was $5*10^5$ CFU/mL prepared in CAMHB (cationic adjusted Mueller–Hinton broth, Sigma-Aldrich) or CAMHB plus 5% of lysed horse blood (TCS Biosciences, Buckingham, UK) in the case of S. pneumoniae. The antimicrobials were 2-fold serially diluted and 50 μL diluted bacterial suspension was added to each well to make the final volume 100 μL. The microtiter plates and the count plates were incubated at 37 °C for 16–20 h without shaking. Growth inhibition was assessed by measuring the $OD_{600}$ using a microplate reader (Tecan Infinity F200). The lowest concentration of the antimicrobials that inhibits visible growth of the indicator strain is identified as the MIC value.

For the synergy test, standard checkerboard broth microdilution assays were conducted to test the synergistic effect of combined antimicrobials using the same condition as before[51,56]. Vancomycin or nisin (compound A) was loaded 2-fold serially diluted at the X axis, while peptides (compound B) were 2-fold serially diluted at the Y axis. The initial concentration of both peptides was their individual MIC. 50 μL fresh bacterial suspension prepared as above was added to the wells already containing the antimicrobial compound mixture to achieve 100 μL final volume per well.

The fractional inhibitory concentration (FIC) indices[57] were calculated using the formula FICI = FICa + FICb = MICac/MICa + MICbc/MICb, to determine whether the combination is additive, synergistic or antagonistic[51,57]. The MICa and MICb is the MIC of compound A or B alone, respectively. MICac is the MIC of compound A in combination with compound B and MICbc is the MIC of compound B when it was combined with compound A. The FIC corresponds to the MIC of a compound in combination with the other compound, divided by the MIC of the compound alone. FICa is FIC of compound A while FICb is FIC of compound B. The FICI was interpreted according to EUCAST[58] as follows: synergistic, FICI ≤ 0.5; additive,0.5 < FICI ≤ 1; indifferent, 1 < FICI < 2; antagonistic, FICI ≥ 2.

For the bactericidal effect testing, the bacteria were inoculated in CAMHB medium at $5 \times 10^5$ CFU/mL and incubated at 37 °C for 3 h in the presence of vancomycin, D-11 alone or their combinations at 10× MIC concentrations (Tables 1 and 3). After that, 50 μL of a $10^3$-fold diluted sample were plated on LB agar[59].

**Plasma stability test**. For plasma stability, the peptides were lyophilized and resuspended in plasma or PBS (as a control) at a final concentration of 3.2 mM. After that, the peptides were incubated with shaking (200 rpm) during 6 h at 37 °C. At time 1, 3, and 6 h, 10 μL samples were tested against *K. pneumoniae* LMG 20218 by a spot-on-lawn test. Reduction in the size of the halo was related to the degradation of the peptide. The test was performed in triplicate.

**Mechanism of action of selected peptides**
*Effect of LPS and $Mg^{2+}$ on the activity and synergy of selected peptides*. The effect of LPS (the major component of the outer-membrane) and $Mg^{2+}$ (divalent cations are involved in an increase of the stability of the outer-membrane and they are essential for its integrity) on the activity of peptide D-11 alone and/or in combination with vancomycin was tested. For this, 21 mM of $MgCl_2$ and 1.0 mg/mL of LPS purified from *E.coli* O55:B5 (Sigma-Aldrich) were added to the CAMHB medium during the MIC and synergistic tests. The condition for the tests and the analysis of the plates were performed as described above for the MIC tests.

*Outer-membrane permeability test*. The integrity of the outer-membranes was analyzed measuring the 1-*N*-phenylnaphthylamine (NPN) uptake after the treatment with the peptide. The outer-membrane acts as a permeability barrier that excludes hydrophobic compounds as NPN. If the outer-membrane is altered, the dye can enter reaching the phospholipid layer which results in a prominent fluorescence[60]. Briefly, the peptides and/or antibiotics were placed in a 96-well plate at 2× the desired final concentration in HEPES buffer plus glucose (GHEPES) (5 mM HEPES, 5 mM glucose). Meantime, the Gram-negative bacteria were grown in LB until an $OD_{600}$ of 1. After that, they were washed twice in GHEPES buffer and 2-fold diluted in the same buffer. NPN was dissolved in acetone and added to the cells at 30 μM final concentration, and after that, the bacteria were distributed in the 96 wells plates previously prepared with the antimicrobials at a final $OD_{600}$ of ~0.25. Immediately the fluorescence for the dye was measured at an excitation/emission range of 350/420 nm every 10 min during 1 h. Polymyxin B was used as a positive control and non-treated cells as negative. All the tests were performed in triplicate.

**Hemolytic activity and cytotoxicity**. Cytotoxicity and the hemolytic activity of peptide D-11 alone as well as in combination with vancomycin were tested by the company Fidelta Ltd (Zagreb, Croatia) and by Wuhan Elabscience Biotechnology Co., Ltd (Wuhan, China). Hemolytic activity was determined with fresh human red blood cells (hRBCs)[28]. The hRBCs were centrifuged (1000×g, 5 min) and the supernatant was discarded. The pellets were washed three times with PBS and diluted in PBS to make the cell stock solution with a cell density of $2*10^8$ cells/mL. The hRBCs suspensions were incubated with vancomycin, the peptide D-11 and their combinations at various concentrations, as indicated. 1% Triton X-100 and PBS were used as controls. The cells were incubated at 37 °C for 1 h and centrifuged at 2000 rpm for 5 min. The supernatant was transferred to a new 96-well plate and hemolysis was monitored by measuring the absorbance at 450 nm ($OD_{450}$). Hemolysis levels were calculated as percentage. % hemolysis = $100 \times (A_s - A_0)/(A_t - A_0)$, where $A_s$, $A_0$, and $A_t$ are the absorbance of the hRBCs suspension in PBS with antimicrobial agents ($A_s$), without antimicrobial agents ($A_0$), and with 1% Triton X-100 ($A_t$). Based on these data, other parameters, such as the HC50 (concentration that causes 50% hemolysis of hRBCs), the MHC (minimal hemolytic concentration that caused 10% hemolysis of hRBCs), and the TI (therapeutic index) were calculated. For the TI calculation (TI = MHC/GM), other factors as the GM (geometric mean of MIC value of all the Gram-negative strains tested) were calculated.

A human embryonic kidney cell line HEK-293 was used to assess the cell viability[61]. 96-well plates were seeded with HEK-293 cells at a concentration of $3*10^5$ cells/well in 100 μL of DMEM growth medium supplemented with 1% NEAA and 10% FBS. Border wells were filled with 100 μL of sterile PBS. Vancomycin and peptide D-11 were added to the cells the next day at various concentrations. ATP levels were measured by adding 50 μL of CellTiter-Glo reagent (Promega, Madison, US) to each well, and luminescence was measured with a SpectraMax i3 microplate reader after 5 minutes of incubation. The potential effect of the tested compounds on cell viability was determined by comparing the signal obtained in the presence of different concentrations of the compounds with those obtained in the control wells. All these tests were performed in triplicate.

**Antimicrobial activity in blood**. Briefly, blood from healthy donors was obtained from Sanquin (certified Dutch organization responsible for meeting the need in healthcare for blood and blood products, https://www.sanquin.nl/) and infected with $10^8$ CFU/mL of *K. pneumoniae* LMG 20218. After that, the infected blood was split in two, and one of the tubes was inoculated with 4 μM of the peptide D-11, and the blood was distributed at 0.5 mL in 2 mL Eppendorf tubes. Vancomycin was added at 16, 8, 4, and 2 μM in the tubes that were incubated at 37 °C for 6 h with gentle shaking. At 1, 2, 4, and 6 h 100 μL samples were taken from each one of the tubes and decimal serially diluted in PBS. Drops of 10 μL of each one of the dilutions were spotted on LB agar medium to follow the development of the infection. The experiment was performed in triplicate. Non-infected blood, infected blood, and blood plus peptide were used as a control.

**In vivo antimicrobial activity test**. As a proof of concept, an in vivo test for the efficacy of the synergy was performed. An overnight *P. aeruginosa* PA14 culture was inoculated into fresh LB medium (100-fold dilution) and grown until an $OD_{600}$ of 1.0. The culture was washed twice and resuspended in 0.9% NaCl to $10^8$ CFU/mL. Four groups of five 6-week-old female BALB/c mice were used. They were anesthetized with 7.5% chloral hydrate by intraperitoneal injection, and then subcutaneously injected with 50 μL of the bacterial suspension at the dorsum of the mice, giving a total infection bacterial number of ~5*$10^6$ CFU per mouse. One hour post-infection, 50 μL of 0.9% NaCl (control group), vancomycin (3 mg/kg), the peptide D-11 (10 mg/kg) and a combination of vancomycin and D-11 at the previous concentrations, respectively, were injected into the infection area. After 24 h, the mice were sacrificed and the abscesses were excised to count the CFU. The bacterial loads were determined by serial dilution and plating. All animal experiments complied with the US and Chinese national guidelines for the use of animals in research. The protocol was approved by the Animal Care and Use Committee of the College of Life Sciences, Nankai University (permission number NK-04-2012). All procedures involving animals were performed after animals were anesthetized. The experimental results were analyzed with the GraphPad Prism software.

**Statistic and reproducibility**. All experiments were performed in replicates as indicated.

**Reporting summary**. Further information on research design is available in the Nature Research Reporting Summary linked to this article.

## Data availability
The data that support the findings of this study are available from the corresponding author on request. Raw data used for Figs. 2 and 3 are shown in Supplementary Data 1.

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

## Acknowledgements

Qian Li was supported by the Chinese Scholarship Council (NO 201306770012). Manuel Montalbán-López was supported by a grant of the EU FP7 Programme SynPeptide. Rubén Cebrián was supported by the NWO-NACTAR program. We thank Prof. Gert Moll for his help with the revision of this manuscript.

## Author contributions

O.P.K. and M.M.-L. conceived and designed the project; Q.L. and R.C. conducted the experiment related to the peptide selection and design, MIC test analysis, synergism combinations, mechanism of action elucidation, toxicity data analysis, and blood test; W.W. and H.R. designed and performed all the mice related experiment and the related writing. Q.L., M.M.-L., R.C.C., and O.P.K. wrote and reviewed the paper. All the authors approved the final paper.

## Competing interests

The authors declare the following competing interests: Part of the results presented in this work have been patented[52]. Each of the authors do not have any interests exceeding $10,000 or 5% equity in a company that has any relation to this work.
