## [Peer Review File · Communications Biology]

Reviewers' comments:

Reviewer #1 (Remarks to the Author):

Thanks for your effort and detailed work.

This article was built on that the outer-membrane-acting peptides were disrupted the outer membrane of the Gram-negative microorganisms to make lipid-II-targeting antibiotics effective. For this reason, candidate molecules (and their improved variants) were tested alone and in combination with vancomycin and nisin. Because the molecule providing the most synergistic activity was unstable in plasma, the d-form of the selected peptide and its reversed form were tested for stability and synergistic activity with lipid-II-targeting antibiotics. D-form of the peptide was preferred for the proper features. To assess the synergistic effect on Gram-negative microorganisms with disrupting the outer membrane, there were three ways. The first of them, synergistic activities on Gram-positive microorganisms were also assessed, and D-11 peptide had no synergistic activity on Gram-positive microorganisms on the contrary of Gram-negative microorganisms. The second of them, Adding LPS and subsequently also Mg²⁺ were decreased synergistic effects of D-11 peptide and lipid-II-targeting antibiotics. And the third one, the disruption on the outer membrane caused by D-11 peptide was tested measuring NPN (1-N-phenyl-naphthylamine) uptake. Cell toxicity and hemolytic activity of D-11 peptide alone and in combination with vancomycin were tested on HEK-293 cells and hRBCs respectively. Antimicrobial activity of D-11 peptide in blood was tested with in vitro infection model. It was evaluated that D-11 peptide was effective and safe with in vitro tests. With a four-arm (Saline, vancomycin alone, D-11 peptide alone and in combination) mouse abscess model, it was shown that vancomycin and D-11 combination might be effective.

I think that this was carried out well and achieved its aim. Results are supported by strong and solitary data.

I believe that data presented will attract attention of the scientific community and have potential to influence future studies.

I would like to suggest authors to further evaluate effectiveness of nisin with in vitro and in vivo tests.

Minor comments:

Line 282 importantly instead of importanly

Line 422 six-week-old instead of six-weeks-old

Reviewer #2 (Remarks to the Author):

The manuscript by Li et al. describes the synergistic effects, against Gram-negative pathogens, of a series of peptides with two Lipid II-binding antibiotics: vancomycin, used systemically to treat infections by Gram-positive pathogens in humans; and nixon, used as a food preservative. While there have been plenty of studies that describe the combinations of LPS-permeabilising agents with different antibiotics that normally cannot cross that barrier, the present study has some merit, especially in the steps for peptide optimisation and in the characterisation of the best combination found. However, there are some issues that the authors need to address.

Main points:

1. Oddly, there were no figure legends in the submitted manuscript. Although the text was sufficiently detailed to understand many of the figures, some details (e.g. Fig. 2BC) are important nonetheless.
2. The rationale for testing synergism with Lipid II-binders is not well explained. In particular, I could not understand why the authors used nisin if their focus was infections in humans.
3. The authors observe best synergy in vitro with a vancomycin to D-11 ratio between 3:1 and

1:1, depending on the species. In the ex-vivo infection, they observe the best effect with a 2:1 ratio. Then, the in vivo experiment is done with a 1:3 ratio of vancomycin to D11. No explanations are provided.

4. The authors should explain why they used *K.pneumoniae* in the ex-vivo and *P.aeruginosa* in the in vivo experiment.

5. The title states of efficient killing. However, the authors do not actually show killing of target pathogens nor its efficiency. In fact, they did not perform any in vitro killing experiments, such as dynamic checkerboards or time-kill curves. The ex-vivo experiment is presented only in a qualitative way. In the in vivo experiment, they used immunocompetent mice, so the decrease in CFU/ml might have resulted from the host clearing the growth-inhibited pathogen.

6. The in vivo experiment was performed by directly injecting the combination into the abscess. As such, it is almost a topical treatment and of limited predictive value about PK and tissue distribution of the combination.

Minor points:

1. there are several instances of typos or awkward English (e.g. lines 211-217). Authors would carefully proof-read their manuscript.

2. Line 40: daptomycin was actually approved for human use in 2003.

3. It would help if MICs were reported in the same table as FICs.

4. Table 4: listing FICIs as "lower than" should be used only when the lowest inhibiting concentrations has not been found, i.e. when the numerator is lower than a given value. If, instead, the MIC of a compound alone is greater than the highest concentration tested, twice that value should be used as the denominator and appropriately explained in the text. For example, the FICI for *E.coli* ATCC 25922 should be reported as 0.321. With the vancomycin FIC being zero, the FICI would still be 0.290.

Rebuttal to Reviewers' comments:

Reviewer #1 (Remarks to the Author):

Thanks for your effort and detailed work.

This article was built on that the outer-membrane-acting peptides were disrupted the outer membrane of the Gram-negative microorganisms to make lipid-II-targeting antibiotics effective. For this reason, candidate molecules (and their improved variants) were tested alone and in combination with vancomycin and nisin. Because the molecule providing the most synergistic activity was unstable in plasma, the d-form of the selected peptide and its reversed form were tested for stability and synergistic activity with lipid-II-targeting antibiotics. D-form of the peptide was preferred for the proper features. To assess the synergistic effect on Gram-negative microorganisms with disrupting the outer membrane, there were three ways. The first of them, synergistic activities on Gram-positive microorganisms were also assessed, and D-11 peptide had no synergistic activity on Gram-positive microorganisms on the contrary of Gram-negative microorganisms. The second of them, Adding LPS and subsequently also Mg²⁺ were decreased synergistic effects of D-11 peptide and lipid-II-targeting antibiotics. And the third one, the disruption on the outer membrane caused by D-11 peptide was tested measuring NPN (1-N-phenyl naphthylamine) uptake. Cell toxicity and hemolytic activity of D-11 peptide alone and in combination with vancomycin were tested on HEK-293 cells and hRBCs respectively. Antimicrobial activity of D-11 peptide in blood was tested with in vitro infection model. It was evaluated that D-11 peptide was effective and safe with in vitro tests. With a four-arm (Saline, vancomycin alone, D-11 peptide alone and in combination) mouse abscess model, it was shown that vancomycin and D-11 combination might be effective.

I think that this was carried out well and achieved its aim. Results are supported by strong and solitary data.

I believe that data presented will attract attention of the scientific community and have potential to influence future studies.

I would like to suggest authors to further evaluate effectiveness of nisin with in vitro and in vivo tests.

Dear reviewer,

Thank you very much for your kind remarks. Although many research groups are working on the design of nisin variants or investigate wild type nisin or other lantibiotics for clinical use, by now, these molecules are not used in clinical practice. In the present work we used nisin because it is a lipid II binding antimicrobial easily available, and with low activity against Gram-negative bacteria due to the outer membrane. So, we considered its use in combination with the peptide to probe the initial hypothesis. From the *in vitro* point of view, the synergistic effect of nisin in combination with the peptides as well as L8 derivatives and D forms of L11 was evaluated (Tables 2, 3 and suppl. Table 1). However, with the current antimicrobial resistance situation, we decided to continue and fully characterize the synergism with clinically relevant Gram-positive antibiotics redirected to fight the most important Gram-negative pathogens. Since vancomycin is already approved for medical use, toxicity studies with the leading peptide screened in this work could attract industrial interest and meet an application in shorter time. Thus we focused our efforts on this point. In addition, the toxicity and effectiveness of lantibiotics, particularly nisin, is a topic that has already been studied in different models. Moreover, recent data on immune system modulation by nisin and anticancer properties are providing a broader picture regarding its applicability.

Field D, Cotter PD, Hill C, Ross RP. Bioengineering Lantibiotics for Therapeutic Success. *Front Microbiol.* 2015;6:1363. doi:10.3389/fmicb.2015.01363

van Heel AJ, Montalban-Lopez M, Kuipers OP. Evaluating the feasibility of lantibiotics as an alternative therapy against bacterial infections in humans. *Expert Opin Drug Metab Toxicol*. 2011;7(6):675-680. doi:10.1517/17425255.2011.573478

de Kwaadsteniet M, Doeschate KT, Dicks LM. Nisin F in the treatment of respiratory tract infections caused by *Staphylococcus aureus*. *Lett Appl Microbiol*. 2009;48(1):65-70. doi:10.1111/j.1472-765X.2008.02488.x

de Kwaadsteniet M, van Reenen CA, Dicks LM. Evaluation of Nisin F in the Treatment of Subcutaneous Skin Infections, as Monitored by Using a Bioluminescent Strain of *Staphylococcus aureus*. *Probiotics Antimicrob Proteins*. 2010;2(2):61-65. doi:10.1007/s12602-009-9017-8

van Staden DA, Brand AM, Endo A, Dicks LM. Nisin F, intraperitoneally injected, may have a stabilizing effect on the bacterial population in the gastro-intestinal tract, as determined in a preliminary study with mice as model. *Lett Appl Microbiol*. 2011;53(2):198-201. doi:10.1111/j.1472-765X.2011.03091.x

Brand AM, de Kwaadsteniet M, Dicks LM. The ability of nisin F to control *Staphylococcus aureus* infection in the peritoneal cavity, as studied in mice. *Lett Appl Microbiol*. 2010;51(6):645-649. doi:10.1111/j.1472-765X.2010.02948.x

We think that the applicability of nisin-vancomycin combinations (or a nisin producing *L. lactis* strain with vancomycin) might be very valuable for use in the gut. However, this topic falls outside the scope of this paper that focuses on Gram-negative pathogens in other human niches

Minor comments:

Line 282 importantly instead of importantly This has been changed in the revised manuscript.

Line 422 six-week-old instead of six-weeks-old This has been changed in the revised manuscript

Reviewer #2 (Remarks to the Author):

The manuscript by Li et al. describes the synergistic effects, against Gram-negative pathogens, of a series of peptides with two Lipid II-binding antibiotics: vancomycin, used systemically to treat infections by Gram-positive pathogens in humans; and nixon, used as a food preservative. While there have been plenty of studies that describe the combinations of LPS-permeabilising agents with different antibiotics that normally cannot cross that barrier, the present study has some merit, especially in the steps for peptide optimisation and in the characterisation of the best combination found. However, there are some issues that the authors need to address.

Main points:

1. Oddly, there were no figure legends in the submitted manuscript. Although the text was sufficiently detailed to understand many of the figures, some details (e.g. Fig. 2BC) are important nonetheless.

Dear reviewer, thank you very much for your comments. We apologize for the mistake during submission; the legends have been included in the revised manuscript.

2. The rationale for testing synergism with Lipid II-binders is not well explained. In particular, I could not understand why the authors used nisin if their focus was infections in humans.

The synthesis of the cell wall is a complex process in bacteria that is essential for cell viability. In fact, many successful antibiotics rely on inhibiting cell wall synthesis. Lipid II is present in all bacteria that have a peptidoglycan cell wall, thus being broadly extended to both Gram-positive and Gram-negative bacteria. Additionally, its biosynthesis and basic structure is common among bacteria and changes are found mainly on the peptide chain attached to the pyrophosphate group. The idea of the project was using lipid II binding drugs, which are well-known to be active against Gram-positive

bacteria, against Gram-negative microorganisms. Since the outer membrane in Gram-negative bacteria is the burden to overcome for lipid II binding drugs, we looked for compounds that could help permeabilizing it. For this reason, we decided to use this target. Next, we considered using two compounds with two different mechanisms of lipid II binding, one on the peptide chain (vancomycin) and the other on the pyrophosphate group (nisin). We have explained this idea better in the revised manuscript in lines 52-55, 68-71, 153-154. Nisin is a well-known, easily available and safe (it has been used for decades in food preservation) lipid II binding antimicrobial in which our group has a broad expertise. For these reasons we used it for the first screening for synergies with different antimicrobial peptides to show proof of concept. After that, the work was focused on the lipid II binding antibiotics currently approved for clinical use, like vancomycin. Nevertheless, nisin and lantibiotic research in general is expanding towards the biomedical field in the last years. In fact, we are currently investigating possible topical/gastrointestinal applications for the combinations with the peptides described.

Breukink E, de Kruijff B. Lipid II as a target for antibiotics. *Nat Rev Drug Discov*. 2006;5(4):321-332. doi:10.1038/nrd2004

Medeiros-Silva J, Jekhmane S, Breukink E, Weingarth M. Towards the Native Binding Modes of Antibiotics that Target Lipid II. *Chembiochem*. 2019;20(14):1731-1738. doi:10.1002/cbic.201800796

Shin JM, Gwak JW, Kamarajan P, Fenno JC, Rickard AH, Kapila YL. Biomedical applications of nisin. *J Appl Microbiol*. 2016;120(6):1449-1465. doi:10.1111/jam.13033

Eckert R. Road to clinical efficacy: challenges and novel strategies for antimicrobial peptide development. *Future Microbiol*. 2011;6(6):635-651. doi:10.2217/fmb.11.27

Li Q, Montalban-Lopez M, Kuipers OP. Increasing the Antimicrobial Activity of Nisin-Based Lantibiotics against Gram-Negative Pathogens. *Appl Environ Microbiol*. 2018;84(12):e00052-18. doi:10.1128/AEM.00052-18

3. The authors observe best synergy in vitro with a vancomycin to D-11 ratio between 3:1 and 1:1, depending on the species. In the ex-vivo infection, they observe the best effect with a 2:1 ratio. Then, the in vivo experiment is done with a 1:3 ratio of vancomycin to D11. No explanations are provided.

First, we apologize for a mistake in the concentration of D-11 used in the *ex vivo* experiment in blood. It was 4 μ M instead 8 μ M for D-11 and 16, 8, 4, and 2 μ M for vancomycin. So the tested ratios for the combinations in the *ex vivo* assay (vacomycin/peptide) were: 4:1, 2:1, 1:1 and 1:2. In the case of the *in vivo* experiment, this relation was selected after screening with other combinations. This screening has been not included in the work because it was performed with a small group of mice per concentration tested. Probably this different optimum is related to different PK properties in blood than *in vitro*.

4. The authors should explain why they used *K.pneumoniae* in the ex-vivo and *P.aeruginosa* in the in vivo experiment.

Initially, *Acinetobacter*, *Klebsiella*, *E. coli* and *Pseudomonas* were tested for their ability to grow in blood. 10^8 CFU/ml of each one were added to the blood and their growth was followed. In the case of *E. coli* and *Acinetobacter* the blood was able to remove the bacteria after about 4h (at least for the tested strains) while *Pseudomonas* and *Klebsiella* were able to grow in these conditions. Between these two, and although the mortality associated to *Pseudomonas* bloodstream infection is higher, the selection of *Klebsiella* was based on the fact that *Enterobacteriaceae* bacteria are the most common Gram-negative bacteria involved in bloodstream infections and *Klebsiella* is highly prevalent. In the case of the *in vivo* experiment, we did not avail of an optimized *Klebsiella* model, therefore we used the *Pseudomonas* one, which also gave convincing results although the *in vitro* data were less good than for *K. pneumoniae*.

We understand the reviewer's concern regarding the use of different strains, but this work is not focused on identifying a treatment for a specific strain, but a proof-of-principle testing the synergistic effect between D11 and vancomycin in different models: *in vitro*, *ex vivo* and *in vivo*. The final idea is to prove that using *in vivo* conditions the treatment is still effective. In the future also other infection models with different bacteria will be used.

5. The title states of efficient killing. However, the authors do not actually show killing of target pathogens nor its efficiency. In fact, they did not perform any *in vitro* killing experiments, such as dynamic checkerboards or time-kill curves. The *ex-vivo* experiment is presented only in a qualitative way.

Vancomycin is a bactericidal antimicrobial so we think that the effect is the cell death. However, to answer this request we have verified this. The obtained data been included in lines 160 to 166 and 357-360. In addition, a Suppl. Figure has been included. These clearly show the killing of the target bacteria.

In the *in vivo* experiment, they used immunocompetent mice, so the decrease in CFU/ml might have resulted from the host clearing the growth-inhibited pathogen.

A control in which only the vehicle was administered was performed. It is included in figure 2C. Perhaps because of the absence of the legend it was not correctly interpreted and we apologize for the inconvenience. In figure 2C, saline is the treatment with saline solution after the infection (infection control). As stated above we convincingly show killing of the bacteria and not just growth inhibition.

6. The *in vivo* experiment was performed by directly injecting the combination into the abscess. As such, it is almost a topical treatment and of limited predictive value about PK and tissue distribution of the combination.

We partially agree with the reviewer on this point. In this work we aimed to obtain evidence of the therapeutical potential of peptides together with lipid II binding molecules as a proof of principle. We show stability and efficacy in *in vitro* and *ex vivo* systems and lack of toxicity in the tests performed. We also think that the widely used subcutaneous administration cannot be regarded as almost topical. A thorough characterization of the *in vivo* toxicity, pharmacokinetic and pharmacodynamics profile of this combination requires setting up appropriate analytical methodologies for D11 that go beyond the scope of this article and will be the topic of follow up work.

Minor points:

1. there are several instances of typos or awkward English (e.g. lines 211-217). Authors would carefully proof-read their manuscript.

We corrected the manuscript carefully to get a better English version.

2. Line 40: daptomycin was actually approved for human use in 2003.

We agree with the reviewer. The sentence is not correct. We have addressed it in the revised manuscript.

3. It would help if MICs were reported in the same table as FICs.

We agree with the reviewer, but Table 2 is already hard to follow and the incorporation of these data could increase the complexity analyzing the results. We prefer to leave the table like this. We have indicated in the legend of table 2 that the MICs alone are listed in Table 1.

4. Table 4: listing FICIs as "lower than" should be used only when the lowest inhibiting concentrations

has not been found, i.e. when the numerator is lower than a given value. If, instead, the MIC of a compound alone is greater than the highest concentration tested, twice that value should be used as the denominator and appropriately explained in the text. For example, the FICI for E.coli ATCC 25922 should be reported as 0.321. With the vancomycin FIC being zero, the FICI would still be 0.290. **The tables have been modified following your indications (also table 2, and suppl. Table 1).**

REVIEWERS' COMMENTS:

Reviewer #1 (Remarks to the Author):

Thanks for your effort and detailed work. Your explanations for my suggestions are enough, and convincing.
Best wishes

Reviewer #2 (Remarks to the Author):

In the revised manuscript and in the rebuttal letter, the authors address most of the issues raised after the original submission.

The revised manuscript is substantially improved and the authors may want to address the following minor points:

- In Table 3, what does "ND" mean? No synergy observed or combinations not tested?
- Table 4: please improved clarity of heading, lines 3-6.
- lines 369-370: medium is missing
- lines 167-173: If I understood correctly the experiment, the authors started with 1×10^5 CFU/ml culture, incubated for 3 hours and then plated 50 μ l of a 1000fold dilution (i.e. 10^2 CFU/ml). Thus, at time zero they plated 5 CFUs only and they cannot state that the combination is bactericidal.
- Line 178: the MIC "of D-11"
- Fig. 2B: it would help the reader if the authors provided additional explanation in the legend and quantitative information in the text. What was the lowest dilution plated? Why is time 0 +D-11 missing? Can the authors on lines 227-233 provide the order of magnitude of the killing effect observed?

REVIEWERS' COMMENTS:

Reviewer #1:

Remarks to the Author:

Thanks for your effort and detailed work. Your explanations for my suggestions are enough, and convincing.

Best wishes

We thank you very for your contribution to the improvement of this paper.

Reviewer #2:

Remarks to the Author:

In the revised manuscript and in the rebuttal letter, the authors address most of the issues raised after the original submission.

The revised manuscript is substantially improved and the authors may want to address the following minor points:

- In Table 3, what does "ND" mean? No synergy observed or combinations not tested?

ND means not tested. This has been included in the revised manuscript.

- Table 4: please improved clarity of heading, lines 3-6.

The heading line has been clarified.

- lines 369-370: medium is missing.

The line has been rephrased

- lines 167-173: If I understood correctly the experiment, the authors started with 1×10^5 CFU/ml culture, incubated for 3 hours and then plated 50 μ l of a 1000fold dilution (i.e. 10^2 CFU/ml). Thus, at time zero they plated 5 CFUs only and they cannot state that the combination is bactericidal.

The amount of cell inoculated was ca. 5×10^5 CFU/mL. After 3 h of incubation if these bacteria are able to grow (with a generation time of 20-30 min for them) we expected more than 10^8 - 10^9 CFU/mL of bacteria. Thus, after 1000-fold dilution and plating more than 10.000 bacteria per plate were expected as we can see in the negative control plates. In the case of a bacteriostatic effect, at least 25-50 bacteria were expected, however, no one was observed for any of the tested strains.

- Line 178: the MIC "of D-11"

The line has been rephrased

- Fig. 2B: it would help the reader if the authors provided additional explanation in the legend and quantitative information in the text. What was the lowest dilution plated? Why is time 0 +D-11

missing? Can the authors on lines 227-233 provide the order of magnitude of the killing effect observed?

The dilution factor has been included in Figure 1. The time 0 +D11 was excluded to simplify the figure. It is the same than for infected blood (first the blood is infected and then divided for D-11 treatment or not). The magnitude order of the killing effect has been included in the lines 222-226 of the revised manuscript.